# Composite GO/Ceramic Membranes Prepared via Chemical Attachment: Characterisation and Gas Permeance Properties

**DOI:** 10.3390/membranes12121181

**Published:** 2022-11-24

**Authors:** Evdokia Galata, Charitomeni M. Veziri, George V. Theodorakopoulos, George Em. Romanos, Evangelia A. Pavlatou

**Affiliations:** 1Laboratory of General Chemistry, School of Chemical Engineering, National Technical University of Athens, Zografou Campus, 9, Iroon Polytechniou Str., Zografou, 15780 Athens, Greece; 2Institute of Nanoscience and Nanotechnology, National Centre of Scientific Research “Demokritos”, Ag. Paraskevi, 15310 Athens, Greece

**Keywords:** GO/composite membranes, chemical attachment, ceramic substrates, gas permeation/separation, linkers, pore structure

## Abstract

Graphene oxide (GO) oligo-layered laminates were self-assembled on porous ceramic substrates via their simple dip-coating into aqueous GO dispersions. To augment the stability of the developed composite GO/ceramic membranes and control the morphology and stacking quality of the formed laminate, short-((3-glycidoxypropyl)trimethoxy silane-GLYMO, (3-aminopropyl)triethoxy silane-APTES), and long-chain (polydopamine-PDA) molecules were involved and examined as interfacial linkers. A comparative study was performed regarding the linker’s capacity to enhance the interfacial adhesion between the ceramic surface and the GO deposit and affect the orientation and assemblage characteristics of the adjacent GO nanosheets that composed the formed oligo-layered laminates. Subsequently, by post-filtrating a GO/H_2_O suspension through the oligo-layered laminate membranes, the respective multi-layered ones have been developed, whereas ethylenediamine (EDA) was used in the suspension as an efficient molecular linker that strongly bonds and interlocks the GO nanosheets. The definition of the best linker and approach was conducted on macroporous α-alumina disks, due to the use of inexpensive raw materials and the ability to fabricate them in the lab with high reproducibility. To validate the concept at a larger scale, while investigating the effect of the porous substrate as regards its micrometer-scale roughness and surface chemistry, specific chemical modifications that yielded membranes with the best gas permeability/selectivity performance were replicated on a commercial single-channel monolith with a ZrO_2_ microfiltration layer. XRD, Raman, ATR, FESEM, and XPS analyses were conducted to study the structural, physicochemical, surface, and morphological properties of the GO/ceramic composite membranes, whereas permeance results of several gases at various temperatures and trans-membrane pressures were interpreted to shed light on the pore structural features. Concerning the short-chain linkers, the obtained results ascertain that GLYMO causes denser and more uniform assembly of GO nanosheets within the oligo-layered laminate. PDA had the same beneficial effect, as it is a macromolecule. Overall, this study shows that the development of gas-separating membranes, by just dipping the linker-modified substrate into the GO suspension, is not straightforward. The application of post-filtration contributed significantly to this target and the quality of the superficially deposited, thick GO laminate depended on this of the chemically attached oligo-layered one.

## 1. Introduction

Graphene oxide (GO) composite membranes have been extensively developed for applications in various fields, such as gas separation [1], water treatment [2], photocatalysis [3], dehydration of ethanol/water mixture via pervaporation [4], etc. Graphene oxide is a two-dimensional (2D) carbon material containing various oxygenated functional groups on its basal planes and at the edges that enable the application of a variety of covalent chemical modification methods with relative simplicity [5]. Of great importance is also the fact that GO can be produced at a large scale by the chemical oxidation of graphite and it is dispersible in water, thus it can be processed with environmentally friendly methods without using organic solvents [6]. All these properties render graphene one of the most attractive materials for bulk applications, such as composites, films, fibers, and membranes. GO membranes can be self-standing [7] and can be supported by polymeric, metal, and ceramic porous substrates. Ceramic membranes are mostly made from alumina, silica, titania, zirconia, or any mixture of these materials. This work is mainly focused on ceramic supports, more specifically Al_2_O_3_ and ZrO_2_, due to their chemical, thermal, and mechanical stability, relatively low-cost, and resistance in corrosive media [8]. They are resistant to high operating temperatures, radioactive/heavily contaminated feeds, and highly reactive environments, where polymeric membranes cannot be used [9].

One of the challenges related to the involvement of GO in membrane technology is the difficulty to strongly and stably adhere on ceramic surfaces, an asset that would allow exploiting the advantages of composite GO/ceramic membranes, especially their endurance at high temperatures and aggressive environments. Stability is a very important benefit for applications, such as wastewater treatment and gas separation. To this end, a promising solution is the chemical modification of the GO and the inert ceramic support with organic molecules that usually contain two kinds of functional groups. The first one is necessary for the attachment of the molecule on the ceramic surface, whereas the second one remains free to interact with the oxygenated functional groups of the GO surface, most of the time through covalent bonding. Thus, endowed with both organic and inorganic properties, these molecules react with both components of the composite, forming durable covalent bonds across the interface.

Various linkers have been studied so far, but the few relevant works suffer from a lack of in-depth elaboration on the possible morphological and pore structural characteristics of the GO/ceramic composite membranes relative to the employed linker. There are still unanswered questions regarding the potentiality of the linker to be densely arranged on the ceramic substrate, in a fashion that the grafted GO sheets possess well-interconnected edges and eventually form a continuous GO layer. In this case, the gas diffusion through the composite membrane is totally controlled by the GO deposit, and the gas molecules access the interlayer space between the stacked GO sheets through structural defects that may exist on the GO sheet. The population of the surface hydroxyl groups on the ceramic substrate (e.g., silanols and aluminols) constitutes another important feature of such an achievement. Equally important are also characteristics, such as the number of functional groups contained in the molecule of the linker along with the possibility that the linker is attached to the ceramic surface by the grafting of either one or all its functional groups. It is also possible that the various ways of linkage affect the orientation of the grafted molecule on the ceramic surface. As such, conformations consisting of a dense arrangement of the molecules with their larger dimension oriented vertically to the substrate’s surface are usually the most promising for concluding a continuous layer of grafted and interlocked GO sheets. Moreover, there may be cases where the linker is introduced into the pores of the substrate bringing together the grafted GO sheets. In this context, a significant reduction in the pore size can be achieved, and the resulting composite membrane may acquire gas separation capacity due to the modification of the substrate’s pores rather than due to the formation of an external GO layer. It is also noteworthy that in most of the recent reports relevant to the development of composite GO/ceramic membranes via the use of linkers, the focal point of the studies pertains to the involvement of the membrane in a specific application and, despite that in most of the cases the works present extraordinary performance, they fail to elaborate and explain the specific structural and morphological characteristics that lead to enhanced flux, selectivity, rejection efficiency, and stability.

For instance, in a recent work, a composite GO/ceramic supported membrane modified with (3-glycidyloxypropyl)trimethoxysilane (GLYMO) was developed, exhibiting a very prominent separation capacity of water from ethanol/water mixtures. Despite that the single gas N_2_ permeation properties were also evaluated, there was not an adequate interpretation of the results in a fashion that would allow for shedding light on the possible pore structural and textural features of the membrane [4]. In a relevant study where the same molecule was employed as a linker, a reduced GO composite membrane was prepared on ceramic pozzolan support with the spin-coating method, while chemical grafting with GLYMO ensured the stability of the deposited laminate. The performance of this membrane for soluble dye removal from the water was elaborated on and found to be very promising [10]. In this case, there were not any discussions about the morphology and structure of the layer along with the grafting conformation of the GO sheets that were responsible for the reported efficiency in rejecting dyes. An example of the use of a different linker is the work of Gu et al. [2], who prepared GO quantum dots (GOQDs) onto ceramic microfiltration membranes for enhanced water permeability with (3-aminopropyl)triethoxysilane (APTES) as a crosslinker. However, there were still missing aspects in the discussion relevant to the optimum way of grafting and the positioning of the GOQDs (into the pores or on the external surface), which were adequate to modify the hydrophilicity of the substrate towards the reported high water flux.

Dopamine is the most ubiquitous linker endowed with two kinds of functional groups (catechol and amino groups), which are appropriate for binding with the inorganic substrate and with the functional groups of GO through non-covalent interactions and covalent coupling. Most of the relevant studies take benefit from the capacity of dopamine to spontaneously oxypolymerize to polydopamine (PDA) under aerobic and alkaline conditions, thus forming a continuous layer and increasing the adhesion of GO on the ceramic support. For instance, Xu et al. [11] studied GO membranes on PDA functionalized ceramic supports for seawater desalination by direct contact membrane distillation, and the water flux through the GO membrane was remarkably high (48.4 kg·m^−2^·h^−1^ at 90 °C), with ion rejections of over 99.7%. The extended stability of the membrane in this work was concluded by the fact that the water flux and ion rejection properties remained unchanged for a period of fourteen days. A more detailed and comprehensive study on the stability of GO/ZrO_2_ composite membranes was conducted by Zhang et al. [12], who followed a hierarchical approach to rationally design and introduce short- and long-chain interlaminar and interfacial molecular bridges that stabilized respectively the GO sheets within the formed laminate and the laminate on the ceramic support. The authors confirmed that PDA is the most adequate molecular linker compared to ethylenediamine and paraphenylenediamine, by concluding notable anti-swelling properties in water and much less invariance of the water permeance and dye rejection properties. Further, aldehyde (glutaraldehyde)-modified chitosan (O=CS) of moderate molecular weight and abundant functional groups was used as a long-chain molecular linker between the PDA-GO laminate and the zirconia substrate, and the respective water permeance results showed noteworthy stability of both water flux and dyes rejection efficiency for twenty-five days. In spite of the merits and the valid conclusions on the type of bonding that dominates the interlaminar interactions between GO and PDA, the interfacial ones between the GO/PDA, the long-chain molecular bridge (O=CH), and the ceramic substrate, this work did not explain issues related to the achieved exceptional interlocking of GO sheets that led to the formation of parallel and intimately stacked GO laminates with prominent dye rejection performance and water flux. Alongside the outstanding flux/separation properties, these membranes presented long-lasting structural stability under cleaning treatment via high-power sonication.

A more detailed study on the tribological, friction, and wear behaviors of polymer composites that employed GO surface-modified inorganic fibers as reinforcing fillers of polyamide 6, was conducted by Wang et al. [13]. Although this work corresponds to a completely different application of GO laminates, the procedures for anchoring the GO laminate on the surface of basalt fibers were very similar to those involved in composite membrane development. The prepared polyamide 6 composites (GO–PDA–BF/PA6) were endowed with increased impact and flexural strength by 13.6% and 12.7%, respectively, as compared to the BF/PA6 composite, mainly due to the large number of active functional groups that have been introduced onto the fibers’ surface and the increase in the surface roughness. Both these assets improved the chemical coupling and mechanical meshing between the fiber and PA6 matrix.

Another work stemming from diverse fields of application, which require high electrical conductivity and optical absorptivity, such as microelectronics, photonics, and light harvesting, is that of Bhowmik et al. [14], who investigated electrically conducting functionalized reduced graphene oxide (FRGO) films on an ordinary glass substrate. Hydrolyzed GLYMO was involved to make covalent ethereal (–C–O–C–) linkages selectively with GO through epoxy polymerization, as well as to anchor GO to the glass substrate through (Si–O–Si) condensation reactions. These hard and durable FRGO films are totally absorbed in the UV–Vis–NIR region of light and electrically conducting.

It is conceivable that the interlaminar covalent bonding within GO laminates along with the interfacial one between the GO laminate and a ceramic surface, comprise customarily used and straightforward ways to develop composites of enhanced stability for a wide range of applications, spanning from membranes to conductive and robust films. However, when the attention is on composite membranes for applications in the gas and liquid phase, the morphology of the GO assembly and the interlocking between adjacent GO nanosheets are equally important with the stability, since they define the diffusion mechanism of the molecules through the membrane, which in turn is definitive for the flux and selectivity properties.

Hence, this work devotes significant effort to unraveling the pore structural features of a great variety of composite GO/ceramic membranes developed with different interfacial linkers. This was achieved by conducting a thorough study of their permeation properties for three gases at different transmembrane pressures and temperatures along with the application of various characterization techniques. Two types of membranes were developed, oligo- and multi-layered GO laminate/ceramic membranes, with the former bearing a few tenths of stacked GO layers and the latter characterized by extended deposition of a few thousand GO layers. Firstly, oligo-layered GO laminate/ceramic composite membranes were prepared by grafting APTES, GLYMO, and PDA on Al_2_O_3_ macroporous disks and ZrO_2_ single-channel microfiltration monoliths and dipping the modified substrates into GO/H_2_O suspensions. Thereafter, to produce the multi-layered membranes, the oligo-layered ones were applied as filters for the vacuum filtration of an EDA containing GO/H_2_O suspension. Innovation is underlined also in this work relevantly to the preparation and testing of the multi-layered composite membranes, with the scope to mend big gaps that compromise the performance of the oligo-layered membranes. To conclude on the pore structural characteristics of the membranes, features such as the extent of permeance reduction compared to the pristine substrate and the dependence of permeance on the pressure and temperature in relation to the adsorptivity of the gas, were interpreted and explained. In addition, the ideal perm-selectivities of several gas pairs were compared to the Knudsen selectivity factor to explicate the dominant diffusion mechanism and elucidate the morphology of the GO assembly and the interlocking at the edges of adjacent GO nanosheets.

Another novelty of this research is that it elaborates and presents for the first time the synthesis, characterization, and gas permeance properties of rGO/ceramic membranes that consist of an rGO/TiO_2_ composite, anchored on the Al_2_O_3_ surface via PDA and further thermally reduced. Though such types of reduced GO/TiO_2_ composites (rGO/TiO_2_) have already been studied in recent works, the membrane deployment was performed solely on polymeric supports [15], while in most of the cases the studies related to the examination of the composite are in its powder form [16]. These membranes were prepared by vacuum filtration of a GO-EDA (ethylenediamine) suspension on the already chemically modified GO/ceramic membranes and, in some cases, they exhibited gas permeance characteristics that could be attributed to microporous diffusion.

## 2. Materials and Methods

### 2.1. Materials

The materials (3-aminopropyl)triethoxysilane 99% (Sigma-Aldrich chemicals, St. Louis, MO, USA), graphene oxide (GO, Abalonyx AS, Forskningsveien, Oslo, Norway), dopamine hydrochloride (Sigma-Aldrich chemicals, St. Louis, MO, USA), tris-Hcl (Sigma-Aldrich chemicals, St. Louis, MO, USA), (3-glycidyloxypropyl)trimethoxysilane ≥ 98% (Fluorochem Ltd., Hadfield, UK), ethylenediamine (EDA) 99% (Alfa Aesar, Haverhill, MA, USA), titanium (IV) butoxide 99% (Acros Organics, Geel, Belgium), and α-alumina powders (Baikalox, CR-6, Baikowski, La Balme-de-Sillingy, France) were employed.

#### 2.1.1. Sol–Gel Synthesis of Graphene Oxide/TiO_2_ Nanocomposite (rGOT) Powder

Ultra-dispersed TiO_2_ nanoparticles were grown on graphene sheets via the sol–gel method. In this context, 20 mg of GO powder (Abalonyx^®^) was dispersed in 200 mL of ethanol and sonicated for 90 min. After 20 min of vigorous stirring, 0.4 mL of NH_3_ was added to the dispersion. The stirring continued for another 30 min followed by the addition of 3 mL of tetrabutyl titanate (TBOT). The sol–gel process had a duration of 24 h. The solids recovered from the solution were washed three times with ethanol. Thermal treatment of the amorphous TiO_2_/GO (GOT) sheets in an Ar atmosphere at 500 °C gave rise to the final nanocomposites. The thermal treatment allowed the transformation of the amorphous TiO_2_ nanoparticles into uniform anatase phase nanoparticles and caused the reduction in GO sheets, leading to the formation of a TiO_2_ nanocrystals/rGO sheets nanocomposite (rGOT) [17].

#### 2.1.2. Preparation of α-Alumina Disks

Custom-made α-alumina disks with a ~2 mm thickness and a 22 mm diameter were employed as supports. The disks were fabricated by pressing commercial α-alumina powder (Baikalox, CR-6) in a custom-made mold with the aid of a hydraulic press (Carver, Inc., Wabash, IN, USA) and sintering at 800 °C for 30 h and further at 1180 °C for 2 h. One side of the disk was polished with SiC sandpaper (Buehler, grit size 600) until no obvious scratch was observed by visual inspection under a light microscope (Olympus, BX61, Tokyo, Japan) with a magnification of ×100 [18].

#### 2.1.3. Preparation of an Oligo-Layered Composite Membrane Consisting of an α-Alumina Disk Modified with PDA-GO

The as-prepared α-alumina disk was first cleaned with H_2_O_2_ (30%) at 60 °C for 10 min to remove possible organic contamination with surface enrichment in the aluminol groups (Al-OH). Due to the high concentration of functional groups in the long-chain PDA linker, it was necessary to further enhance the population of the aluminol groups on the α-alumina surface with the target to achieve the most possible anchoring sites between PDA and the ceramic disk. Accordingly, the cleaned disk was subjected to a quite aggressive basic hydrolysis treatment that took place in a NaOH solution of pH = 9.5 at 70 °C.

For anchoring the PDA on the surface of the fully hydroxylated a-Al_2_O_3_ disk, 0.1 g Tris–HCl (pH = 8.5) was initially dissolved in 75 mL deionized (DI) water, followed by the addition of dopamine (2 mg/mL). The disk was immersed in the solution and left under stirring at 20 °C for 20 h, so that dopamine was oxypolymerised to the PDA forming a deposited layer on the ceramic surface [11,19]. Subsequently, the PDA-coated, a-Al_2_O_3_ disk was dipped into a 1 mg/mL GO/H_2_O solution.

#### 2.1.4. Preparation of an Oligo-Layered Composite Membrane Consisting of an α-Alumina Disk Modified with PDA-rGOT

The above-described procedure for depositing PDA on a-Al_2_O_3_ disks [11,19] and synthesizing the amorphous GOT nanocomposites was replicated up to the point where amorphous GOT was recovered from the sol and washed repeatedly with ethanol. The PDA-treated ceramic support was then dip-coated into a 1 mg/mL GOT/H_2_O solution. Subsequently, by thermal treating the produced ceramic composite under an Ar atmosphere at 500 °C [17], the development of the final composite membrane was achieved. The membrane held a deposited laminate consisting of anatase TiO_2_ nanocrystals/rGO sheets nanocomposite.

#### 2.1.5. Preparation of an Oligo-Layered Composite Membrane Consisting of an α-Alumina Disk Modified with APTES-GO

Compared to the long-chain PDA, the short-chain molecular linker (APTES) possesses a lower number of functional groups (ethoxy groups), which are responsible for the covalent bonding with the aluminol groups existing on the substrate’s surface. Hence, there was no need here for the application of aggressive basic hydrolysis treatments. As such, to both clean the substrate and increase the population of the surface aluminols, the alpha alumina disks were first boiled in 30% hydrogen peroxide for 10 min and then dried at 150 °C for 2 h. After that, the ceramic membranes were soaked in ethanol for 10 min and submerged into a solution composed of 15 μL APTES in 10 mL of ethanol at 25 °C for 90 min. Then, the membranes were again thoroughly cleaned with copious amounts of ethanol to remove the unreacted APTES and dried at 60 °C [2]. In the final stage, the membranes were dip-coated into a 0.2 mg/mL GO/H_2_O solution and heated at 100 °C for 2 h to allow the formation of amide linkages between the oxygenated groups of GO (carboxylic, epoxy) and the primary amine groups (NH_2_) of APTES (Figure 1).

#### 2.1.6. Preparation of an Oligo-Layered Composite Membrane Consisting of an α-Alumina Disk Modified with GLYMO-GO

GLYMO is also a short-chain molecular linker exhibiting a small number of methoxy groups that can be covalently anchored on the alumina surface. Hence, similar to the case of APTES, boiling in 30% hydrogen peroxide for 10 min was enough to introduce hydroxyl groups onto the surface of the ceramic disks, which were then dried at 150 °C for 2 h. Subsequently, the hydroxylated disks were immersed into a GLYMO/absolute ethanol solution (2% *v*/*v*) for 30 min at 40 °C. The silylated disks were heated for 4 h at 110 °C and dip-coated into a GO aqueous solution (1 mg/mL) for 1 min. Finally, the disks were dried at 50 °C [4] (Figure 2).

#### 2.1.7. Multi-Layered Composite Membrane Prepared by GO Vacuum Filtration on an Oligo-Layered APTES-GO Substrate (Al_2_O_3_ APTES GO-F)

GO (Abalonyx^®^) was dispersed in DI water at a concentration of 0.1 mg/mL followed by ultrasonication for 30 min. Then, EDA (5 wt.%) was added into the aqueous GO dispersion and the solution was subjected to 15 min ultrasonication and 16 h mechanical stirring at room temperature with the target to enhance its homogeneity. The filtration of the homogeneous solution through the oligo-layered APTES-GO membrane was carried out with a syringe without agitation [20].

#### 2.1.8. Multi-Layered Membrane Prepared by GO Vacuum Filtration on an Oligo-Layered GLYMO-GO Substrate (Al_2_O_3_ GLYMO GO-F)

The procedure for the preparation of the EDA containing GO/H_2_O suspension (0.1 mg/mL) was similar to the one described in the development of the multi-layered composite membrane on the APTES-GO deposited substrate. The only difference here is that the filtration was performed using a peristaltic pump, which circulated the suspension from a glass container equipped with magnetic stirring, to the feed side of the oligo-layered GLYMO-GO membrane, and from the retentate outlet back to the glass container, while the permeate side was maintained under high vacuum. The target was to conduct the filtration under conditions of continuous flow and agitation, which limit the generation of air bubbles and conclude in the formation of a homogeneously coated surface. Normally, all filtration procedures lasted 48 h [20].

### 2.2. Filtration System

To develop the multi-layered membranes on top of the oligo-layered APTES-GO and GLYMO-GO substrates, a custom-made filtration system was used which operated in the cross-flow filtration mode. As shown in Figure 3, the system consists of five main components. A turbomolecular pump supported by a rotary vacuum pump is involved to keep the permeate side of the membrane under high vacuum conditions (10^−3^ mbar) during the entire filtration process. A vacuum trap connected in the line between the pumping system and the permeate outlet port of the membrane cell ensures that the filtrate does not contaminate the internals of the turbomolecular pump and the oil of the rotary pump. A glass vessel with magnetic stirring, a peristaltic pump (a range of 1–60 mL/min), and the inlet/outlet ports (feed/retentate) of the membrane cell are the main components of a closed circulation circuit for the GO suspension. The peristaltic pump abstracts the suspension from the glass vessel and conveys it to the feed port of the membrane cell. The GO suspension sweeps the feed side of the membrane tangentially and from the retentate outlet port, it is dispatched back to the glass vessel. Usually, the filtration process lasted for about 48 h, or until the consumption of all the contained suspension in the glass vessel. In the following step, the membrane was placed in an oven and dried at 50 °C, whereas cross-linking between EDA and GO was promoted due to the elevated temperature. The drying process was carried out gradually to avoid the GO-EDA layer peel-off from the ceramic support.

### 2.3. Permeability Measurements

Single gas permeance measurements for the as-prepared GO/ceramic composite membranes have been conducted in a home-made stainless-steel permeability rig, operating in the dead-end mode (Figure 1). The measurements were performed at three temperatures (25, 60, and 100 °C) and various trans-membrane pressures. Details of the measurement method and device can be found in a previous publication [7].

### 2.4. Characterization

In order to study the structural, morphological, and surface chemistry properties of the GO/ceramic composite membranes, various techniques were applied, such as XRD, micro-Raman, ATR-FTIR, FESEM, and XPS. For the XRD analysis (D8 Advance, Bruker, Germany), the measurements were performed at 2-theta angles within the range of 5° to 80° and with a scanning rate of 0.03°/min employing Cu-Kα radiation (λ = 1.5418 Å) at a voltage of 30 kV and a current of 15 mA. The confocal microscope Raman apparatus (inVia, Renishaw, Wotton-under-Edge, Gloucestershire, UK) used two excitation sources, that of a solid-state laser (λ = 532 nm) and that of a high power near-infrared (NIR) diode laser (λ = 785 nm). Raman measurements were performed at room temperature in a backscattering configuration and the laser beam was focused onto the samples by means of an ×20 short distance magnification lens with low excitation power, in order to secure low laser heating of the samples. The frequency shifts were calibrated by an internal Si reference. Two to three spots were analyzed for each sample. The exposure time was 10 s, with 2–10 accumulations. An Attenuated Total Reflectance (Brucker FTTR Spectrometer Alpha II, which features a monolithic diamond crystal) was used to define the surface chemistry of the GO/ceramic composite membranes. The morphological features were examined by using a Field Emission SEM (FESEM, JSM-7401F, JEOL, Tokyo, Japan). Each sample was pretreated by gold sputtering prior to the FESEM observation. The elemental composition on the surface of the composite membranes and the binding states of the elements were measured by X-ray photoelectron spectroscopy (Leybold SPECS). The photoemission experiments were carried out in an ultra-high vacuum system (UHV) with a base pressure of 1 × 10^−9^ mbar using an un-monochromatized MgKα line at 1253.6 eV. The XPS core level spectra were analyzed using a fitting routine, which can decompose each spectrum into individual mixed Gaussian–Lorentzian peaks after a Shirley background subtraction.

## 3. Results

### 3.1. Raman Analysis

Figure 2a depicts the Raman spectrum of an as-prepared Al_2_O_3_ disk operating as a membrane substrate, which is characterized by typical bands of α-Al_2_O_3_ located at ~378, 431, 453, 581, and 752 cm^−1^ and assigned to E_g_ (external) and at 416 (most intense) and 645 cm^−1^ to A_1g_ modes [21]. In the case of GO ceramic composite membranes, the Raman spectra are mainly dominated by the Raman D and G bands of carbon-based materials in the region of 1000–2000 cm^−1^ (see Figure 2b).

The D and G bands in the Raman spectra of all examined samples (Figure 2b) are associated with structural features of GO-modified membranes. The D band, for example, is associated with structural defects of graphene and its intensity is amplified as the degree of oxidation increases. The ratio between the intensities of these peaks (I_D_/I_G_) is also related to structural variations in the GO and is indicative of the level of oxidation/reduction. High I_D_/I_G_ values are attributed to the enhanced concentration of oxygenated functional groups on the surface of graphene, while low values indicate either low concentration or reduction in these groups with the later not being necessarily true, since in most of the cases the GO reduction generates higher sp^3^ hybridization related to the removal of functional groups and formation of defects [22]. In fact, the D band represents the out-of-plane vibration attributed to the structural defects. On the other hand, the G band is a result of in-plane vibration related to the sp^2^ graphitic domains (non-oxidized regions). Thus, when the D band is higher, it is an indication that sp^2^ bonds are broken, which, in turn, means more sp^3^ carbon is present [23]. In this context, the ratio of the intensities between the D and G bands (I_D_/I_G_) depends on the level of disorder, i.e., sp^3^ defects within the sp^2^ hybridized graphene in the functionalized-GO. The I_D_/I_G_ ratios of GO and the oligo-layered composite membranes Al_2_O_3_ GLYMO GO, Al_2_O_3_ PDA GO, and Al_2_O_3_ APTES GO were 1.01, 1.14, 1.21, and 1.61, respectively (Table 1).

The enhanced I_D_/I_G_ ratio manifests the prevalence of chemical bonding as the dominant mechanism of the linker’s anchoring to GO. This kind of chemical attachment between the functional groups of GO and these of the silanes and PDA generates more sp^3^ carbon within the sp^2^ carbon network of graphene, resulting in a higher I_D_/I_G_ ratio compared to pure GO. Hence, in the case of APTES which exhibits the highest I_D_/I_G_ ratio among oligo-layered membranes, Moon et al. proposed that the electrons of the amine group (lone electron pair) attack the electrophilic carbon of the epoxide group on GO, causing ring opening and resulting in a negative charge on the remaining oxygen and a positive charge on the bonded nitrogen (ammonium). The oxygen abstracts hydrogen from the ammonium resulting in an alcohol and an amine group. This epoxide-to-alcohol transformation is a reduction process which, as mentioned above [22], is associated with the observed significant enhancement of the corresponding I_D_/I_G_ ratio. Hence, it could be stated that the varying degree of enhancement on the I_D_/I_G_ ratio caused by the three different linkers is indicative of differences in the mode, extent, and strength of grafting. For instance, the difference in the I_D_/I_G_ ratio in the case of GLYMO and APTES indicates that APTES is attached to the Al_2_O_3_ surface through condensation reactions between the surface aluminol groups and its ethoxy groups, while the anchoring to GO is achieved through nucleophilic attack of its amine groups to the epoxide rings on GO (see Figure 1). On the other hand, GLYMO possibly follows a different mechanism when acting as a molecular bridge between GO and alumina, which at this stage it is speculated (and further confirmed by the interpretation of the XPS results) to take place through a condensation reaction of one or two of its methoxy groups with the aluminol groups on Al_2_O_3_, along with breakage of the C–O–C linkage and bonding with the α-Al_2_O_3_ surface at a second aluminol site. Hence, the further anchoring of GO takes place on unreacted methoxy groups of GLYMO, which are attacked by the hydroxyl groups of the GO surface (condensation reactions) (see Figure 2). On this basis, according to the analysis relatively to the I_D_/I_G_ ratio, APTES could be regarded as the more effective linker in GO laminate stabilization.

Nevertheless, it is also important to examine the Raman shift of the position of D and G bands caused by the anchoring on GO (Appendix A), along with the results obtained from other analytical techniques (see the XRD and XPS results discussed in the following Section 3.2 and Section 3.3) to draw reliable conclusions on the effectiveness of the linkers. For instance, the blue-shift of the G band position confirms the assimilation of a linker’s moiety in the GO framework, implying enhanced attachment to the graphene layer [24]. PDA was the only linker that caused a blue-shift of 3 cm^−1^ in the wavenumber of the G band (APTES and GLYMO caused a red-shift, (Appendix A)) and, therefore, it can be considered as the one exhibiting the higher binding capacity for GO. This conclusion also converges with the following discussed interpretations of the XPS and XRD results.

An additional remark relevant to the observed shifts in wavenumber caused by the linkers (Appendix A) is that the D band for Al_2_O_3_ APTES GO underwent a significant red-shift of 32 cm^−1^ after GO’s functionalization with the aminosilane. This is owing to grafting that provokes a positive inductive effect on the primary amine functional groups attached to GO, which act as strongly nucleophilic electron donor substituents [25]. On the contrary, the grafting of GLYMO, which contains a highly electrophilic epoxy group, caused a blue-shift of 7 cm^−1^, whereas the attachment with PDA, a polymer with aromatic nitrogenated groups donating a lone pair to their aromatic systems and, therefore, being less reactive, did not have a significant effect on the wavenumber of the D band.

Overall, the interpretation of the Raman results concerning the oligo-layered composite membranes reveals that, although the grafting with APTES generates a higher level of structural disorder in GO, APTES seems to be a less effective linker as compared to GLYMO and PDA.

Regarding the multi-layered laminate composite membranes Al_2_O_3_ GLYMO GO-F and Al_2_O_3_ APTES GO-F, developed by the post-filtration of an EDA containing aqueous dispersion of GO through the Al_2_O_3_ GLYMO GO and Al_2_O_3_ APTES GO oligo-layered composites, the I_D_/I_G_ ratios appear readily enhanced, reaching values of 2.43 and 2.12, respectively (Table 1). This was an expectable characteristic for the multi-layered membranes since their Raman signal is mainly assigned to the thick GO laminate consisting of GO sheets interlocked with EDA. Considering the enhanced I_D_/I_G_ ratio caused by APTES, it was apparent that an even stronger nucleophile, such as EDA, having two electron donor groups and interlocking two adjacent GO sheets, would cause a more profound enhancement of the I_D_/I_G_ value. It should be also noted that the high degree of disorder on the extra deposited GO nanosheets implies that reacting sites located on GO, mainly epoxides, are attacked by the primary amine groups of EDA. Hence, the formation of additional defects, which is reflected by the augment of the I_D_/I_G_ ratio, provides direct evidence that the target of interlocking the GO nanosheets within the thick laminate has been achieved [26]. In the case of Al_2_O_3_ PDA rGOT, a slight decrease in the I_D_/I_G_ ratio was observed (I_D_/I_G_ = 0.9) for the anchored laminate composed of the rGO-TiO_2_ nanocomposite. The lower I_D_/I_G_ ratio compared to pure GO can be attributed to the decrease in the distance between domains of carbon atoms with sp^2^ hybridization caused by the reduction of sp^3^ to sp^2^ carbon during the thermal treatment [27]. However, this result cannot be confirmed since the anatase phase TiO_2_ peaks are absent in the Raman spectrum, possibly due to the decomposition of PDA during the thermal treatment, resulting in the delamination of the formed layer. Indeed, the gas permeance performance of this membrane (discussed in Section 4.1.1) illustrates that the amount of the deposited nanocomposite was very low.

### 3.2. XRD Spectra of the Oligo-Layered and Multi-Layered GO Laminate/Ceramic Composite Membranes

XRD was used in order to investigate the crystallinity of the different GO/ceramic composite membranes. Figure 3 presents all the diffraction diagrams recorded for the as-produced membranes and Figure 4 presents the diffraction angle area of the GO’s Bragg reflections for all chemically modified membranes. The peaks at about 10° correspond to GO and the other peaks are representative of α-Al_2_O_3_ (JCPDS 71-1683) [21].

The average crystallite size was calculated by using Scherrer’s equation, d = 0.89 λ/β cos θ, where d is the average crystalline size, 0.89 is the Scherrer’s constant, λ is the X-ray wavelength, θ is the diffraction angle, and β is the FWHM (full-width-half-maximum). This is calculated for the main peak of GO at 2θ = 10°. The average crystallite size of all the thin films is on the nano-scale. In detail, it was estimated at 9.01 nm, 4.19 nm, 4.06 nm, 5.84, and 4.67 nm for Al_2_O_3_ GLYMO-GO, Al_2_O_3_ APTES-GO-F, Al_2_O_3_ PDA-GO, Al_2_O_3_ GLYMO-GO-F, and Al_2_O_3_ APTES-GO membranes, respectively.

Moreover, shifts in the Bragg reflections of GO were observed due to the different molecular linkers used to stabilize the GO laminate on the surface of the ceramic substrate. The multi-layered laminate membranes prepared by the post-filtration method presented two peaks at ~8° and 24°. This is indicative of the two types of laminate attached to these membranes, the oligo-layered and the multi-layered ones, and confirms that the anchoring of adjacent GO nanosheets by EDA during the post-filtration method was successful [28].

The d-distance of the GO stacks on the oligo-layered and multi-layered membranes was calculated by Bragg’s Law equation. At this point, we should stress that having knowledge of the d-distance is very important in supporting the further interpretation of gas permeance results and elucidating the pore structural characteristics of the developed membranes (Section 4.1). In the ideal case of a perfectly assembled GO laminate that consists of densely stacked and interlocked GO nanosheets, the pore structure comprises both the structural defects (holes) existing on the GO layer and the interlayer spacing between adjacent GO nanosheets. In this context, the mechanism of gas permeation through the deposited GO laminate encompasses the passage of gas molecules through the structural defects and their further diffusion through the interlayer galleries. Hence, the interlayer spacing (d-distance) between GO sheets constitutes one of the most important characteristics of the GO laminate and, according to the size calculated for the oligo-layered membranes Al_2_O_3_ GLYMO-GO, Al_2_O_3_ APTES-GO, and Al_2_O_3_ PDA-GO, it can be concluded that micropore diffusion would be the dominant mechanism of gas permeation in case the formed laminates were characterized by a dense and free of defects assembly of GO nanosheets. Specifically, the d-distance was estimated at 8.45 Å, 8.5 Å, and 9.57 Å, for the oligo-layered Al_2_O_3_ GLYMO-GO, Al_2_O_3_ APTES-GO, and Al_2_O_3_ PDA-GO membranes, respectively. The preparation of this type of composite membrane starts with the anchoring of the molecular linkers (GLYMO, APTES, and PDA) on the ceramic surface and proceeds with the immersion of the ceramic disk into the GO/H_2_O suspension. Provided that the bonding is strong and resistant to hydrolysis, there is no possibility for the covalently bonded linkers to be also intercalated into the GO interlayer galleries of the deposited oligo-layered laminate, and their role is limited to acting as a linking medium between the bottom GO layer of the laminate and the ceramic surface. Thereupon, the next few layers of GO are kept close to each other solely through physical van der Waals interactions (π-π attraction) and their d-spacing should depend on the surface properties of GO, especially on the degree of its functionalization with oxygenated groups [29]. Even though, it is interesting that despite using the same batch of GO in all synthetic processes, the few GO layers on the Al_2_O_3_ PDA-GO composite membrane exhibited distinctively higher d-spacing (9.6 Å) compared to the other two (8.5 Å). This notable difference is attributed to the molecular structure of the linker and the strength of its interaction (chemical bonding) with the first layer of the laminate. Explanatively, APTES and GLYMO possess active functional groups (aliphatic amine and epoxy), with the capacity to strongly bind with GO. Particularly APTES, having an aliphatic amine group as an electron donor, constitutes a strong nucleophile with high electron density. However, both APTES and GLYMO, due to their short chain length, convey high electrostatic repulsion between the negatively charged Al_2_O_3_ surface and the negatively charged GO surface of the bottom layer. In this context, the first GO layer is loosely coupled to the α-Al_2_O_3_ surface and the van der Waals interactions between the first and subsequent GO layers within the oligo-layered laminate become significant, giving rise to small d-spacing (8.5 Å). Contrarily to the short-chain linkers, PDA features reactive alicyclic amine groups and is endowed with a molecular configuration of high steric hindrance that bears less electrostatic repulsion between the similarly charged surfaces of α-Al_2_O_3_ and the first GO layer. Thus, the stronger binding of the first GO layer on the surface causes the van der Waals interactions with the succeeding layers to become less significant, leading to much higher d-spacing (9.6 Å).

A noteworthy difference is also observed in the d-spacing of the oligo-layered and multi-layered membranes. The multi-layered Al_2_O_3_ GLYMO-GO-F and Al_2_O_3_ APTES-GO-F exhibited d-spacing of 10.39 Å and 12.49 Å, respectively. As observed in the cross-section images of all samples obtained by SEM analysis (Figure 5 and Figure 6), the thickness of the GO-deposited laminate in the multilayered composite membranes is much higher than that of the oligo-layered ones. Consequently, the information obtained by the XRD analysis is mostly relevant to the interlocked EDA layers of GO, which are deposited during the filtration process, whereas the first few GO layers that were attached during the preparation of the Al_2_O_3_ APTES-GO and Al_2_O_3_ GLYMO-GO substrates are not of sufficient quantity to contribute to the XRD spectrum. Therefore, we figured out that the higher d-spacing of the multilayered composite membranes was attributed to the intercalation of EDA into GO layers. Moreover, EDA is a linker of very short chain length and, in spite of forming a strong nucleophile with the capacity to covalently bond successive GO nanosheets (through nucleophilic addition reactions with the epoxy groups of GO) and keep them very close to each other, it also brings up very high electrostatic repulsion from succeeding GO nanosheets being at a very close distance. As such, there is a counterbalancing effect that results in the observed higher d-spacings.

### 3.3. Interpretation of XPS Results

XPS survey spectra were recorded to explore the surface elemental compositions, nature, and chemical states of the functional groups of all the composite samples, excluding the multi-layered ones (Appendix A). It should be noticed that XPS analysis was not attainable on the as-prepared composite GO/ceramic membrane disks for reasons related to the type of samples, which are inappropriate for introduction into the measurement chamber of the spectrophotometer. However, to make possible the acquisition of valuable information from this powerful analytical technique, the composite GO/ceramic samples were also developed on α-Al_2_O_3_ powder. In this context, the previously described synthetic methodologies of all the oligo-layered composite membranes were replicated with the only difference being that α-Al_2_O_3_ in the form of powder, instead of a shaped disk, was used and dispersed in the solutions of the linkers and the GO/H_2_O suspensions. The amount of α-Al_2_O_3_ powder involved in each synthesis was 2 g, which was equal to the amount used to prepare one disk. As such, since the compression of α-Al_2_O_3_ powder and the further sintering of the shaped disk at 1180 °C for 2 h caused a significant reduction in the specific surface area; the powder samples exhibited a much lower surface concentration of anchored linker and GO compared to the sintered disks. Therefore, the interpretation of XPS results can provide qualitative information associated with the anchoring of the linker and its possible conformation on the α-Al_2_O_3_ surface, but it is not suitable for drawing quantitative conclusions on the extent of deposition, especially for GO.

Hence, in the XPS spectrum of the GO surface, only C 1s and O 1s peaks are observed (Table 2 and Table 3), while in the XPS spectrum of the Al_2_O_3_ PDA-GO surface, N 1s was observed at binding energy of about 400 eV. This peak corresponds to primary amine (NH_2_), and aromatic/tertiary amine (N–C) functional groups, respectively, as presented in Table 2. The primary amine is linked to the anchored dopamine, and the tertiary amine corresponds to the tautomers of 5,6-dihydroxyindole and 5,6-indolequinone [30]. The existence of the tautomers indicates that dopamine has undergone the stages of oxypolymerization on the α-Al_2_O_3_ surface including oxidation, intermolecular cyclization, and rearrangement, so the functionalization with PDA was successful. Al_2_O_3_ APTES-GO contains the elements C, O, Al, and traces of Si. As explained above, the amount of Si is very small because of the very high surface area of α-Al_2_O_3_ powder, which results in a very low surface concentration of APTES.

On the other hand, GLYMO seems to be much more effectively attached on the ceramic surface as compared to APTES. Despite that both short-chain molecular linkers have one Si atom in their structure, the XPS analysis of the GLYMO-GO/α-Al_2_O_3_ composite provided a Si atomic concentration of 3.5% when Si for APTES was marginally detected.

Based on the Si atomic content and considering the atomic concentration of all the other elements present on the surface of the GLYMO-GO/α-Al_2_O_3_ composite, it was possible to draw conclusions on the possible conformation of GLYMO’s anchoring and its specific arrangement on the surface. GLYMO holds one Si atom, nine C atoms, and five O atoms. If the grafting on the α-Al_2_O_3_ surface takes place via condensation reactions between the three methoxy groups of GLYMO and the surface aluminol groups, then the anchored GLYMO loses three C atoms and three O atoms (removed as methanol). Consequently, the C % and O % atomic concentrations associated with the attachment of GLYMO (for 3.5% Si) must be 3.5 × 6/1 = 21% C, and 3.5 × 2/1 = 7% O. Subtracting the 7% O content from the total oxygen concentration of the composite (51.1%, Table 3) gives 44.1% O, a percentage that encompasses the oxygen of α-Al_2_O_3_ and the oxygen of the attached GO. Since the Al atomic concentration is 30.1% (Table 3), the O which corresponds to the α-Al_2_O_3_ substrate is 30.1 × 3/2 = 45.5%. The two numbers are almost identical, indicating that the initial assumption of GLYMO being mostly attached on the surface with all its methoxy groups may be correct. Nevertheless, a prerequisite for this conformation is that the area spotted by XPS is free from GO. This is because the surface of GO is fully oxygenated and its presence would give rise to a much higher atomic concentration of oxygen. Although the absence of GO can be further evidenced by the percent of atomic concentration of C attributed to GLYMO (21% as calculated), which is higher than the C percent of atomic concentration on the entire composite (15.4%, Table 3), the presence of -COOH groups on the sample (Table 2) is a feature that can be only assigned to the existence of GO. Therefore, another more plausible conformation is that GLYMO lies tangentially on the α-alumina surface and is attached through the condensation reaction of two methoxy groups along with the breakage of the C-O-C linkage and the bonding with the α-Al_2_O_3_ surface at a second site. This type of binding keeps the molecular linker flat on the surface and in this case the anchored molecule remains with four C atoms and one O atom. Conducting similar calculations as above, the carbon and oxygen atomic concentrations attributed to the attachment of GLYMO are 14% and 3.5%, respectively. Therefore, there remains an excessive amount of C atomic concentration of 1.4% (15.4–14), which is assigned to the presence of GO, as well as a 2.1% atomic concentration of O (51.1–45.5–3.5) attributed to the oxygenated functional groups of GO. The higher probability of this configuration is further supported by the following point. APTES can only be attached to the ceramic surface through condensation reactions between its ethoxy groups and the aluminol groups. If GLYMO was to be anchored in the same way as APTES, then its Si content would not be detectable by XPS. Therefore, a different anchoring mechanism for GLYMO is behind its higher effectiveness for grafting.

Another notifiable feature for discussion rises by examining the ratio of C-C sp^2^ to C-C sp^3^ hybridization of the carbon content in the different samples analyzed by XPS. It can be seen (Table 2) that the ratio of the percent relative concentration of carbon in the sp^2^ hybridization state over the percent relative concentration of carbon in the sp^3^ hybridization state varies from very high (GO is 2.82) to very low (APTES-GO/Al_2_O_3_ is 0.61). Considering that C atoms in aliphatic chains of organic molecules are in the sp^3^ hybridization state (the case of APTES and GLYMO) while aromatic rings consisting of sp^2^ hybridized carbon (the case of PDA), and that GO possesses mostly sp^2^ hybridized carbon, this variation can be indicative for the extent of GO deposition on the several substrates. Thus, for APTES-GO/Al_2_O_3_ the C% (sp^2^)/C% (sp^3^), the ratio is 0.61, meaning that the carbon detected by XPS is mostly associated with the C atoms in the aliphatic chain of APTES and that GO is poorly attached on the α-Al_2_O_3_/APTES surface. Moreover, the α-Al_2_O_3_/GLYMO and the α-Al_2_O_3_/PDA surfaces must exhibit almost the same efficiency in binding GO, with C% (sp^2^)/C% (sp^3^) ratios of 1.31 and 1.35, respectively. In fact, the slightly higher ratio for PDA is attributed to the sp^2^ carbon in the aromatic rings. Regarding the PDA-GO/α-Al_2_O_3_ composite, based on the N atomic content and considering the atomic concentration of all the other elements present on the surface of the composite, it was possible to draw conclusions on the possible configuration of PDA’s anchoring and the extent of GO deposition. The PDA unit possesses one N atom, eight C atoms, and two O atoms. Assuming PDA’s grafting on the α-Al_2_O_3_ surface through the two hydroxyl groups of the aromatic ring, it can be stated that there is no loss of O atoms due to anchoring. Thus, for a 5.5% atomic concentration of N, as measured by XPS, the PDA attachment on the α-Al_2_O_3_ surface brings up a C% concentration of 44% (5.5 × 8/1 = 44%) and an O% concentration of 11% (5.5 × 2/1 = 11%). The large difference between the carbon content attributed to PDA (44%) and the total C content of the composite (54.4%, Table 3), is indicative of the significant amount of GO that is deposited on the surface. Similarly, conducting the respective calculations for the O content of PDA and α-Al_2_O_3_, an excessive amount of 0.7% O is determined, which is associated with the functional groups of GO.

### 3.4. Morphological Features of the Oligo-Layered and Multi-Layered GO Laminate/Al_2_O_3_ Composite Membranes

The surface and cross-sectional morphological characteristics of the custom-made oligo-layered and multi-layered GO laminate/ceramic composite membranes, developed on α-alumina disks of 0.2 μm pore size, were assessed by FESEM (Figure 5). GO nanosheets seem to be interconnected to each other and form a thin layer on the modified ceramic supports (Figure 5a,b,d). The thickness of GO nanosheets, which are anchored on the Al_2_O_3_ surface, is related to the linker that is used for the modification and is in good agreement with the d-spacing calculated by XRD. No visible cracks, pinholes, or other defects are found on the composite membranes, while the large particles existing on the top surface of the formed GO laminate had been transferred from the ceramic substrate during the sample preparation for SEM analysis, which took place by cross-cutting the membrane with a diamond wheel (Figure 5a). GO nanosheets tend to conform to the surficial structure of substrates, resulting in thin GO selective layers snugly attached on top of the substrates [31]. In Figure 5b, a conformal morphology of the membrane surface with apparent wavy-wrinkles can be observed clearly due to the thickness of the GO layer. In an oligo-layer laminate Al_2_O_3_ PDA GO membrane, the cross-section view of the membrane with a thickness of ~200 nm is tightly attached to the surface of the Al_2_O_3_ due to the high adhesive ability of PDA to GO nanosheets (Figure 5a). The thickness of GO nanosheets with GLYMO as a linker is ~38 nm (Figure 5b), while APTES as a linker is ~100 nm. In multi-layered laminate filtration, the thickness of Al_2_O_3_ GLYMO GO-F is ~4700 nm (Figure 5c). The surface morphology of the multi-layer membrane exhibited a uniform and densely packed structure that can be associated with the directional flow caused by filtration during the manufacturing of the membrane and the addition of EDA that acts as a linker between the GO sheets.

Figure 6 depicts the morphology of chemical modifications that was replicated in a commercial monolith having a top-layer of ZrO_2_. The cross-section image of ZrO_2_ GLYMO GO-F (Figure 6a) shows that multi-layer GO has been successfully deposited on the ZrO_2_ surface. A top view image (Figure 6b) of an oligo-layer ZrO_2_ PDA GO membrane depicts that the GO layer is uniformly coated on the support. The GO layer adheres quite well with the ceramic support, further confirming that the membrane was successfully prepared by the dip-coating method on PDA-treated ceramic support. The surface of the ZrO_2_ composite membrane is smoother compared to the Al_2_O_3_ ones, due to possibly smaller pore size and the preparation method applied in commercially available membranes that involves either a slurry of fine powders or a colloidal sol prepared by hydrolyzation of metal oxides in an excess of water. As referred to above, GO nanosheets tend to conform to the surficial structure of substrates, resulting in thin GO selective layers snugly attached on top of the substrates.

### 3.5. ATR-FTIR Results

Figure 7 depicts ATR spectra for all modified (b, c, d, and e) and unmodified samples (a). Figure 7a presents the IR spectrum of Al_2_O_3_ APTES GO flat membrane that exhibits peaks at around 541 cm^−1^, 633 cm^−1^, 1059 cm^−1^, 1245 cm^−1^, 1592 cm^−1^, 2932 cm^−1^, and 3138 cm^−1^. Strong absorption bands at 541 cm^−1^ can be assigned to the stretching vibration of Al–O bonds [32]. The peaks at 3138 cm^−1^ are attributed to –OH stretching, at 1592 cm^−1^ to the sp^2^ C=C and conjugated C=O stretching. N–H stretching vibrations are present at 3138 cm^−1^ and C–H stretching bands in the region 2932 cm^−1^. As evidence of covalent functionalization, the appearance of absorption peaks can be seen at 1059 cm^−1^ and 1245 cm^−1^, indicating Si–O–C and Si–O–Si, respectively [25].

The IR spectrum of Al_2_O_3_ GLYMO GO (Figure 7b) with peaks at 555 cm^−1^ and 633 cm^−1^ correspond to Al–O bonds, and the peaks at 1053 cm^−1^, 1382 cm^−1^, 1621 cm^−1^, and 3208 cm^−1^ reveals the presence of C–O and C–O–C from epoxy or ether, C=O, and –OH groups, respectively [4].

Figure 7a shows the corresponding spectra of the Al_2_O_3_ substrate and pure GO. Strong absorption bands of Al_2_O_3_ blank at 577 cm^−1^ can be assigned to the stretching vibration of the Al–O bond [32]. The spectrum of GO is characterized by a broad and strong peak at 3175 cm^−1^, which is attributed to the O–H stretching vibration of the OH– moieties. Furthermore, the stretching vibration of C=O moieties appears at 1712 cm^−1^, while the remaining graphitic domains (C=C) stretching vibrations are shown at 1613 cm^−1^. In addition, C–O–H bending vibration due to COOH groups is presented at 1374 cm^−1^. The peaks at 1122 cm^−1^ and 1035 cm^−1^ represent the C–OH stretching vibration of the hydroxide domains and the stretching vibration of the C–O–C groups, respectively [33].

Figure 7d presents the ATR-FTIR spectrum of the Al_2_O_3_ PDA GO flat membrane. Peaks at 530 cm^−1^ and 631 cm^−1^ correspond to Al–O bonds as referred. Peaks at 1720 cm^−1^, 1080 cm^−1^, and 1561 cm^−1^ correspond to C=O stretching vibrations, C–O, and (CON–H), respectively [11]. In Figure 7e, the spectrum of Al_2_O_3_ GLYMO GO-F membrane was presented, and characteristic absorption peaks of a GO-modified membrane were detected at wavenumbers of 1072 cm^−1^, 1421 cm^−1^, and 1640 cm^−1^, corresponding to the stretching vibrations of C–O in alkoxy group, C–O in carboxyl group, and C=C in the aromatic ring, respectively. Meanwhile, the peak at 2918 cm^−1^ corresponds to the C–H stretching vibrations of the methylene group from the bonded EDA. In addition, a peak observed at 1522 cm^−1^ along with another band at 1640 cm^−1^ are attributed to the bending of N–H and the stretching of C=O on the secondary amide group, respectively [20].

## 4. Discussion

### 4.1. Gas Permeance Measurements

#### 4.1.1. Gas Permeance Results at Non-Adsorbing Conditions

The experimental study on the gas diffusion through the GO/ceramic composite membranes in concomitance with the survey relative to their pore structural features, initiated by measuring the He permeance (Pe_He_) at 100 °C and examining various transmembrane pressures (TMP) up to 1.2 bar. The rationale behind the selection of these specific conditions as a starting point of our experimental campaign is that by examining a non-adsorbable gas at elevated temperature, diffusion mechanisms attributed to gas adsorption (surface diffusion) are limited. Hence, the focus stays solely on the Knudsen and Hagen-Poiseuille mechanisms of gas transport through membranes, along with the transition regime (slip flow) between them. In this context, the micropore diffusion process (or molecular sieving process), which describes the motion of gas molecules through pores (micropores) of a dimension close to the kinetic diameter of gases, is excluded at this stage of discussion.

According to the model developed by Weber, the general expression of the gas permeability through a porous medium in the mixed Poiseuille–Knudsen flow regime is described by the following equation [34]:(1)$$P^{\prime }=\frac{J\ell }{U_{c\;}\Delta C}=\frac{4\varepsilon \delta d_{h}}{3\tau }{\left(\frac{2RT}{\pi M}\right)}^{\frac{1}{2}}+\frac{\varepsilon P_{\mu }d_{h}^{2}}{8\eta \;\tau }$$
where, $P^{\prime }\;\left(m^{2}\cdot s^{-1}\right)$ the permeability of the porous membrane, ℓ (m)  the thickness of the membrane, $J\;\left(\mathrm{mol}\cdot s^{-1}\right)$ the gas flux through the membrane, $U_{c}\;\left(m^{2}\right)$ the surface of the porous membrane, $\Delta C\;\left(\mathrm{mol}\cdot m^{-3}\right)$ the gas concentration difference across the membrane, ε the porosity of the membrane, τ the tortuosity factor defining the real length (τ·ℓ) of the capillaries within the membrane, $d_{h}\;\left(m\right)$ the hydraulic diameter of the capillaries, η (Pa·s) the gas viscosity, $P_{\mu }\;\left(\mathrm{Pa}\right)$ the mean pressure, T (K) the temperature of the gas, $M\;\left(\mathrm{kg}\cdot {\mathrm{mol}}^{-1}\right)$ the molecular weight of the gas, $R=8.314\;\left(J\cdot {\mathrm{mol}}^{-1}\cdot K^{-1}\right)$ the gas constant, and δ, a dimensionless coefficient accounting for the contribution of Knudsen flow and slip flow, which depends on the Knudsen number ($K_{n}=\frac{\lambda }{d_{h}}$), where λ (*m*) is the mean free path of the gas molecule. The graphical representation of $P^{\prime }\;f\left(P_{\mu }\right)$ is a curve that presents a minimum for large Knudsen numbers ($K_{n}\geq 10)$ and converges asymptotically to a linear form of the type $P^{\prime }=A+BP_{\mu }$, with *A* corresponding to the mixed contribution of Knudsen and Slip flow and *B* corresponding to the Poiseuille flow.

According to the kinetic theory of gases, the mean free path (λ) of He at 100 °C and pressures up to 550 mbar (the mean gas pressure is considered here, calculated as the average of gas pressures at both sides of the membrane) can vary from λ = 3.4 μm (at 50 mbar) down to λ = 310 nm at a mean pressure of 550 mbar.

Furthermore, when the mean free path of gas molecules is much larger than the pore dimension $d_{h}$ ($d_{h}<0.1\;\lambda$, i.e., for large Knudsen number), the second term in Equation (1) becomes zero and the coefficient δ takes the value of 1. Under these conditions, Knudsen diffusion takes place (permeation evolves solely via collisions of the gas molecules with the pore walls) and the permeance through the membrane is independent of the transmembrane (TMP) pressure (Equation (1)), whereas for larger pore dimensions and at the same conditions of temperature and pressure, the transition to Poiseuille flow commences via the slip flow mechanism, which comes first into play (for 0.1 λ < $d_{h}$ < 100 λ), and the permeance correlates linearly with TMP.

The plots presented in Figure 8 illustrate the dependence of Pe_He_ on the TMP at 100 °C for all the developed membranes, including the bare Al_2_O_3_ and ZrO_2_ substrates. It is apparent that the Pe_He_ of both substrates exhibits slip-to-Poiseuille flow features. Hence, considering that the lower TMPs examined were in the range of 50–100 mbar and that during the measurement the permeate side of the membranes was kept under vacuum conditions, it was possible to estimate that the pore dimension of both substrates is larger than 340 nm. Focusing now on the prepared GO/Al_2_O_3_ composite membranes (Figure 8a), it can be concluded that only the ones bearing a multi-layered laminate (prepared by post-filtration), together with the oligo-layered Al_2_O_3_ GLYMO GO, and Al_2_O_3_ PDA GO, presented purely Knudsen flow characteristics up to a TMP of 1100 mbar (mean pressure of 550 mbar). Accordingly, their pore dimension is less than 30 nm. Although the setup of the involved permeability rig was not adequate for measuring the permeance at higher TMPs, a missing asset that would allow us to gain a clearer view of how narrow the pores are of the developed membranes. There are two noteworthy issues for discussion here that underline the importance of the obtained results. The first issue is the agreement between the conclusions drawn from the permeance and spectroscopic experiments. Interpreting the results of the Raman analysis in conjunction with XPS, we identified APTES as the less effective molecular bridge for the development of oligo-layered membranes. This is further confirmed by the finding that Al_2_O_3_ APTES GO exhibits Poiseuille flow within the entire pressure range applied in the experiments (Figure 8a), which is indicative of a very loose conformation of the oligo-layered stacking along with the existence of large gaps between the neighboring GO nanosheets. The same holds for Al_2_O_3_ PDA rGOT. It should be mentioned that the expected anatase phase TiO_2_ bands were not detected in the Raman spectrum of this membrane. Moreover, the Al_2_O_3_ PDA membrane that was used as a substrate for the chemical attachment of the rGOT composite exhibited a He permeance of 2.6 × 10^−6^ mol/m^2^/s/Pa, while the permeance of the derived Al_2_O_3_ PDA rGOT membrane under the same conditions (TMP of 100 mbar, 373 K) was 2.9 × 10^−6^ mol/m^2^/s/Pa (see also Appendix A). The increase in permeance as compared to the substrate is an indication that during the thermal treatment at 500 °C (the temperature required to transform amorphous TiO_2_ nanoparticles to crystalline anatase phase), the PDA decomposed leading to partial detachment of the deposited rGOT composite. Hence, both the Raman analysis and Pe_HE_ results indicate the loose structural features and low concentration of the deposited rGOT composite, caused due to decomposition of PDA during the thermal treatment that resulted in the delamination of the formed laminate.

The second issue relates to the extent of permeance reduction in the GO/Al_2_O_3_ composite membranes as compared to the bare Al_2_O_3_ disk. First to note is that all the composite membranes endowed with Knudsen flow characteristics exhibit, in parallel, a Pe_He_ reduction of more than one order of magnitude as compared to the bare substrate. This holds also for the ones developed on the ZrO_2_ monolith (Figure 8b). The second remark is that a high Pe_He_ value does not suggest that the developed oligo-layered GO laminate is loose and full of gaps. Hence, we turn the focus to the Al_2_O_3_ PDA GO sample, as an example of a high flux membrane (high Pe_He_) with Knudsen flow characteristics, and we aspire to explain its distinguishing features. The key issue is that up to this point, the comparison was based on the permeance (Pe) and not on the permeability factor (P′). In seeking to compare two membranes in terms of their pore structural characteristics, which in the case of the composite GO/ceramic membranes reflect the coherence of the formed GO laminate, the key performance indicator must be the P′. The reason is that P′ is an inherent property of the material, independent of the laminate thickness, whereas the gas permeance (Pe) describes the permeability over the length of the flow path. Hence, the SEM cross-section images of Figure 5a,c allowed us to estimate the laminate’s thickness on the oligo-layered Al_2_O_3_ PDA GO membrane at about 38 nm, whereas the thickness of the multi-layered GO laminate on Al_2_O_3_ GLYMO GO-F was found to be 4.7 μm. Considering that the d-spacing of the GO layers on Al_2_O_3_ PDA GO is 0.957 nm (from the XRD analysis) and that the thickness of one graphene sheet is 0.335 nm, a total number of twenty-four GO layers could be defined for Al_2_O_3_ PDA GO, while when conducting the same calculations for Al_2_O_3_ GLYMO GO-F it was found that a total number of about three thousand GO layers composed the thick laminate deposit. Accordingly, by multiplying the Pe_He_ values of Al_2_O_3_ PDA GO and Al_2_O_3_ GLYMO GO-F membranes at 100 °C and a TMP of 250 mbar (1.8 × 10^−6^ and 3.7 × 10^−7^ mol·m^−2^·s^−1^·Pa^−1^, respectively, seen in Table 4) with the defined thicknesses, the respective permeability factors were 6.9 × 10^−14^ and 1.7 × 10^−12^ (mol·m·m^−2^·s^−1^·Pa^−1^). Conclusively, the oligo-layered GO laminate on Al_2_O_3_ PDA GO exerts a much higher resistance to the flow of He as compared to the multi-layered deposit on Al_2_O_3_ GLYMO GO-F, meaning that the GO nanosheets are more uniformly stacked on the ceramic substrate and in intimate proximity each other so that the distances between their edges are minimized. We followed the same procedure for the respective composite membranes developed on the ZrO_2_ monolith (Figure 8b) and the estimated permeability factors were two orders of magnitude lower, indicating that the higher smoothness and possibly the higher concentration of hydroxyl groups on the ZrO_2_ surface impart a significantly beneficial effect on the conformation of the GO nanosheets’ assembly. The overall results are appended in Table 4.

Although both membranes developed on the ZrO_2_ monolith pose a significant restriction to the passage of He molecules, it is not reasonable to consider this property as an indicator of enhanced gas separation capacity, or as evidence that the formed oligo-layered or multi-layered laminates are composed of perfectly stacked and interlocked GO nanosheets. In addition, in the latter case, the gas transport through the membranes would result in diffusion through micropores, which is an activated process (in many cases mentioned as activated Knudsen diffusion), having the distinctive feature of increasing permeance with the rise of temperature. Actually, none of the membranes developed in this work exhibited such an increasing trend and the dependence of gas permeance on temperature indicates that Knudsen diffusion describes better the mechanism of gas transport through their pore structure. Before going further to the discussion on the permeation properties of other adsorbable gases, where surface and Knudsen contributions to permeance must be elaborated, in the following paragraph we focus on several differences of the Pe_He_ dependence on temperature that were pointed out for some of the GO/ceramic membranes, and we try to relate them with distinct pore structural features.

#### 4.1.2. Dependence of Pe_He_ on Temperature

In Figure 9, the temperature dependence of Pe_He_ for all the GO/ceramic composite membranes is presented. To ensure that the observed correlations are not affected by interferences due to Poiseuille flow, which is already evidenced to take place for some of the membranes (Section 4.1.1), the plotted values of permeance vs. temperature were obtained at the same TMP, e.g., 250 mbar for the Al_2_O_3_ and 300 mbar for the ZrO_2_ deposited samples.

The gas permeance via Knudsen diffusion varies with the square root of the inverse temperature ratio. Although this seems to be contradictory to what is described by the Knudsen term in Equation (1), it can be explained based on the units in which the permeance is expressed. As such, when the unit for permeance is $\left(m\cdot s^{-1}\right)$, the dependence on temperature is exactly as described in Equation (1). However, when the permeance is measured in $\left(\mathrm{mol}\cdot m^{-2}\cdot s^{-1}\cdot {\mathrm{Pa}}^{-1}\right)$, then (ΔC) must be multiplied by $\left(RT_{m}\right)$, where $\left(T_{m\;}\;\left(K\right)\right)$ is the temperature of the membrane. Hence, Equation (1) is transformed to:(2)$$P^{\prime }=\frac{J\ell }{U_{c\;}\Delta C\;R\;T_{m}}=\frac{4\varepsilon \delta d_{h}}{3\tau }{\left(\frac{2T}{\pi RMT_{m}^{2}}\right)}^{\frac{1}{2}}+\frac{\varepsilon P_{\mu }d_{h}^{2}}{8\eta \;\tau RT_{m}}$$

Equation (2) shows the variation with the square root of the inverse temperature ratio in the Knudsen regime, especially when the temperatures of the gas and the membrane are the same.

From the plots depicted in Figure 9, it becomes clear that solely the bare substrates and the oligo-layered GO laminate membranes developed using APTES as a molecular bridge, including the Al_2_O_3_ PDA rGOT sample, follow this general trend. The correlation of Pe_HE_ with temperature for the rest of the developed membranes deviates significantly from the Knudsen-defined trend and the deviation becomes higher as the temperature drops from 333 K to 298 K. These features appear in a more perceptive way in Table 5. Hence, all the multi-layered composite membranes, independently on the substrate and the molecular linker, along with the oligo-layered membranes developed by anchoring the GO laminate with GLYMO and PDA, exhibit a remarkable deviation that can only be attributed to the contribution of surface flow to the overall diffusion mechanism.

This asset unveils that He molecules nesting inside the interlayer galleries of stacked GO nanosheets are readily adsorbed due to the enhanced attractive potential imposed by the proximity of the opposite GO surfaces. The d-spacings defined by XRD are in the range of 8.5 to 12.5 Å. Consequently, the dimension of the GO galleries classifies them as being of microporous size. Diffusion in micropores (micropore diffusion) proceeds via three mechanisms, which include surface diffusion, gas translation diffusion, and configurational diffusion. The dominance of each mechanism depends on the ratio of the pore diameter over the kinetic diameter of the gas molecules, with the configurational regime coming into effect for a ratio of 1, whereas the surface diffusion mechanism dominates when the ratio is higher than around 1.25. In our case, the ratio of the interlayer GO galleries over the kinetic diameter of He is in the range of 2–3. It can be, therefore, concluded that the multi-layered membranes and the oligo-layered ones developed with GLYMO, and PDA hold quite extended regions onto their surface, where the deposited GO laminate is characterized by a perfectly organized assembly of stacked GO nanosheets. The gas molecules can permeate through these regions only through their passage from structural imperfections of the GO nanosheet and further diffusion through the microporous interlayer galleries. Hence, He adsorption into the microporous galleries is readily enhanced, leading to the enhanced contribution of surface diffusion at 25 °C. These are reflected by the abrupt rise of permeance at 25 °C (Figure 9a,b, Table 5). Since the largest deviation of Pe_HE_ at 25 °C was observed for the multi-layered laminate/ceramic composite membranes, ZrO_2_ GLYMO GO-F, Al_2_O_3_ GLYMO GO-F, and Al_2_O_3_ APTES GO-F it can be stated that these membranes hold GO laminates of enhanced integrity. This means that if the deposited laminate is composed of domains characterized by loose and dense structural assemblies of the GO nanosheets, the dense domains dominate in covering the membrane surface. Further conclusions relative to the gas separation capacity of the membranes and the possibility that the formed laminate totally controls the diffusion of gases are provided in the following section.

#### 4.1.3. Elaboration of Gas Separation Performance in Conjunction with the Knudsen Selectivity Ratio

The permeances of He, CO_2,_ and CH_4_ through the membranes were measured at 298.15 K, 333.15 K, and 373.15 K, and various TMPs up to 1.2 bar. The entire set of results is appended in Appendix A.

In Figure 10, a comparison between the developed membranes regarding their capacity to separate He (the gas with the smaller kinetic diameter) from CO_2_ and CH_4_ at 373.15 and 298.15 Κ, and TMPs of 100, 250, and 500 mbar is depicted, along with the respective Knudsen selectivity ratio.

Presenting the results in this fashion facilitates the elaboration of data toward defining the best performing composite membranes, characterized by deposited GO laminates of enhanced integrity. In this respect, the interpretation of the Pe_HE_ results (Section 4.1. and Section 4.1.2) has hitherto concluded that all the developed in this work composite GO/ceramic membranes carry a chemically anchored and continuous GO laminate, which sprawls on their top surface through alternating domains of high- and low-density assemblies of GO. The dense domains are the ones consisting of well-stacked and interlocked GO nanosheets and are endowed with microporous diffusion characteristics that are mostly expressed with the surface diffusion mechanism due to the relatively large size of the inter-layer GO galleries as compared to the kinetic diameter of the examined gases. The looser domains are characterized by the existence of voids between adjacent GO stacks. Nevertheless, for some of the membranes, these voids (empty spaces) are of mesopore size. Therefore, depending on the size of the voids the gas diffusion mechanism through, the low-density domains can span from Knudsen to Poiseuille flow, whereas the surface diffusion mechanism on the pristine graphene part of the GO nanosheet has a minor contribution to the overall flux.

From this perspective, the membranes of enhanced integrity and performance must bear deposited layers whose structure is mostly occupied by dense domains.

The results presented in Figure 10 show that the post-filtration derived multi-layered laminate membranes, despite their distinguishing Pe_HE_ dependence on temperature, which was indicative of the dominance of microporous domains within the laminate’s structure (Section 4.1.2), exhibited Pe_HE_/Pe_CO2_ selectivity that was lower than the Knudsen separation factor (Knudsen He/CO_2_ = 3.32). In addition, the respective oligo-layered laminate membranes presented higher Pe_HE_/Pe_CO2_ selectivity and, amongst them, Al_2_O_3_ APTES GO and ZrO_2_ GLYMO GO were endowed with a Pe_HE_/Pe_CO2_ separation performance that overpassed the Knudsen separation factor at the specific conditions of high temperature (373 K) and a low TMP (100 mbar) (Figure 10).

Endeavoring to explain these contradicting results, it is necessary to define and elaborate reasons that could justify He/CO_2_ separation performance as higher than expected by Knudsen diffusion. As such, since He has a smaller kinetic diameter than CO_2_; the prevailing cause for a membrane to manifest enhanced He/CO_2_ separation performance is to own a purely microporous structure, where gas transport takes place through a molecular sieving process. However, this cannot be the case for our membranes since, subject to the same conditions, their He/CH_4_ separation performance should be much higher than what is predicted by Knudsen diffusion (CH_4_ has a larger kinetic diameter than CO_2_). Nevertheless, both oligo-layered membranes (Al_2_O_3_ APTES GO, and ZrO_2_ GLYMO GO) that presented higher Pe_HE_/Pe_CO2_ selectivity as compared to the Knudsen selectivity factor, at the same time exhibited Pe_HE_/Pe_CH4_ selectivity values (Figure 10a) which were less than the predicted ones by Knudsen diffusion (Knudsen He/CH_4_ = 2).

Another frequent reason that renders a GO laminate membrane capable to separate He from CO_2_ with a higher performance than the one achieved by Knudsen diffusion, relates to the strong adsorptivity of CO_2_ on the oxygenated groups emanating from the GO surface. As such, the intercalated CO_2_ molecules are significantly hindered in moving through the interlayer galleries since they must overcome a high energy barrier (being desorbed and then adsorbed again to the next oxygen group—this is similar to diffusion taking place through a hopping mechanism between adsorption sites). In this context, Pe_HE_/Pe_CO2_ values greater than expected for Knudsen diffusion indicate that GO is attached in a way that most of the oxygenated functional groups are free and exposed towards the core of the pores to interact with CO_2_ molecules. This is exactly what happens in the oligo-layered membranes. Apart from the first GO layer, which is anchored on the ceramic surface through chemical bonding between the oxygenated groups and those of the organic linker, the next few layers (25–50, Table 4) that compose the deposited laminate the oxygenated groups remain intact and are available to interact with CO_2_ (there is no linker in-between the succeeding GO layers). In addition, the above results confirm that in the case of the oligo-layered membranes, gas diffusion mostly takes place through the looser domains that dominate the laminate structure. In this context, surface diffusion on the pristine part of the GO nanosheets, an asset that would enhance the CO_2_ permeance, has a very low contribution to the overall flux. The reason is that the CO_2_ adsorption on the pure graphene surface is moderate because of the weak attractive potential generated between opposite GO surfaces being at a long distance from each other.

Contrariwise, the deficient Pe_HE_/Pe_CO2_ separation capacity of the multi-layered laminate membranes is not indicative of the existence of bulky empty spaces within the laminate but rather of the higher degree of its integrity. Explanatively, the gas permeation through the multi-layered composite membranes is totally controlled by the extra-deposited and quite thick GO laminate. As confirmed by the results of the Raman analysis, most of the oxygenated functional groups of the GO nanosheets that compose the thick laminate are attacked by the nucleophilic amine groups of EDA. As such, they are not available to interact with the diffusing, through the interlayer galleries, CO_2_ molecules. Furthermore, apart from the vanishment of the hindering effect of the GO’s functional groups, CO_2_ permeation benefits from the enhancement effect of surface diffusion in micropores. Large amounts of CO_2_ are adsorbed on the purely graphitic areas of the GO nanosheets under the attractive potential generated by the proximity of the opposite GO layers. As a result, the contribution of surface diffusion to the overall permeation mechanism is significantly enhanced. In addition, examining the histograms of Figure 10 from top to down (an increase in TMP) and from left to right (a decrease in temperature), it can be concluded that the Pe_HE_/Pe_CO2_ separation capacity of the multi-layered laminate membranes decreases with the increase in TMP, as expected because of the higher CO_2_ adsorption at elevated TMPs, whereas the trend vs. temperature is not straightforward. To provide a clearer view of this issue, in Figure 11 we depict the dependence of Pe_HE_/Pe_CO2_ permselectivity on temperature. In most of the cases, the Pe_HE_/Pe_CO2_ permselectivity presents a minimum at about 333 K. Considering that the Pe_HE_ declines continuously as the temperature increases (Figure 9), the observed minimum signifies that the permeance of CO_2_ (Pe_CO2_) is significantly augmented at the specific temperature. The fortification of the CO_2_ flux up to a certain temperature confirms that the dominant mode of gas diffusion in the multi-layered membranes is surface diffusion into micropores. Explanatively, the mechanism of surface diffusion into micropores is affected by temperature in two ways. Firstly, surface diffusion is an activated mechanism, and the gas diffusivity is amplified, as the temperature increases. On the other hand, the adsorbed amount declines so that the adsorbed concentration gradient over the membrane’s thickness passes through a maximum. The combination of these two effects produces a maximum in permeance with respect to temperature. Although the co-existence of rare low-density domains significantly suppresses the intensity of the CO_2_ flux enlargement, the above-described events of minimum Pe_HE_/Pe_CO2_ selectivity and maximum CO_2_ flux with respect to temperature constitute further evidence that the gas permeation in this type of multi-layered laminate composite GO/ceramic membranes takes place mostly through the interlayer galleries of dense laminate domains. Moreover, the fact that the minimum in the Pe_HE_/Pe_CO2_ separation capacity with respect to temperature appeared for all the membranes and at all the examined TMPs, signifies that the CO_2_ adsorption on the plain graphene domains of the GO nanosheets within the deposited laminate is significant even at very low pressures (Figure 11a–c). Since the CO_2_ adsorption at low pressures can be significant solely in micropores, this asset implies that the larger segment of the deposited laminate consists of dense domains of high structural integrity.

Hereafter, with the target to determine the best performance among the three multilayered membranes, we shifted our focus to their Pe_HE_/Pe_CH4_ separation capacity (Figure 10a–f). Distinguishing features of CH_4_ as compared to CO_2_ is its larger kinetic diameter (3.8 Å compared to 3.3 Å for CO_2_) and its lower adsorptivity.

In addition, contrary to what happens with CO_2_, the CH_4_ permeation is not hindered by the presence of oxygenated functional groups, since CH_4_ has not any specific affinity for them. As such, CH_4_ permeation is mostly contributed by the surface diffusion mechanism taking place on the plain graphene domains. Therefore, the He/CH_4_ ratio must be lower than expected by the Knudsen mechanism. This holds for all the membranes developed in this work (both oligo-layered and multi-layered) (Figure 10), with the exception of one case, this of the multi-layered ZrO_2_ GLYMO GO-F membrane, which at low TMP (100 mbar) and high temperature (373 K) had achieved a Pe_HE_/Pe_CH4_ separation factor of 2.34 (Figure 10a). Considering that CH_4_ is less adsorbable than CO_2_, it was expected that the Pe_HE_/Pe_CH4_ separation capacity would be closer to the Knudsen selectivity as compared to the Pe_HE_/Pe_CO2_ separation capacity. Indeed, from the data presented in the plots of Figure 10, it could be calculated that depending on the conditions, the Pe_HE_/Pe_CO2_ separation capacity of all the multi-layered membranes varied between 68% to 89% of the respective Knudsen separation factor, whereas the Pe_HE_/Pe_CH4_ separation capacity was in the range of 68% to 117%, with the higher percentages achieved by ZrO_2_ GLYMO GO-F. This result points out that the pore structure conformation of ZrO_2_ GLYMO GO-F is capable to inhibit the diffusion of the bulkier CH_4_ molecules.

#### 4.1.4. Structural Features of the Membranes Elucidated by Gas Permeance in Relation to the Organic Linker and the Pore Structure of the Substrate

Having the target to elucidate whether a molecular linker attached to the internal (pore) surface of the ceramic substrate has the capacity to attract and anchor the GO nanosheets, trapping them inside the pores of the substrate, we have also evaluated the gas permeance properties of a linker-modified substrate, before the attachment of GO. The selected linker was PDA because of its macromolecular structure that significantly hinders possible penetration into the substrate’s pores. Since the two other linkers used in this work (APTES and GLYMO) are small molecules, it was definite that they would be eventually grafted on the pore walls without creating any blocking effect. Hence, the concept behind the selection of PDA was that we wanted to first study an oligo-layered laminate GO/Al_2_O_3_ composite membrane, where both the linker and the anchored GO laminate are deposited solely on the external surface, and further use it as a reference case for comparison with the other oligo-layered membranes.

When PDA is attached to the Al_2_O_3_ surface, the Pe_He_ drops to a large extent (e.g., at a TMP of 250 mbar, it drops from 3.5 × 10^−6^ to 2.6 × 10^−6^ mol/m^2^/s/Pa) (Figure 12a). This is an indication that the bulky PDA chains that lie on the external Al_2_O_3_ surface hinder significantly the permeation of He. In parallel, when dopamine undergoes oxidative polymerization to PDA, the nucleophilic nitrogen atom of its primary amine group reacts with one carbon of the catechol ring forming a five-membered ring with the nitrogen enclosed as a heteroatom. As a result, PDA is rich in secondary amine groups, which have the capacity to bind CO_2_ molecules. It can be seen (Figure 12b) that the Al_2_O_3_ PDA membrane exhibits a very high He/CO_2_ permeance ratio at 373 K and TMP of 100 mbar, which goes beyond the Knudsen selectivity factor. This is because the secondary amine groups strongly adsorb and delay the passage of CO_2_. When GO is further attached to the PDA-modified Al_2_O_3_ membrane, the He permeance declines to 1.8 × 10^−6^ mol/m^2^/s/Pa and this is evidence that GO nanosheets are transferred from the solution and stabilized on the membrane surface, thus generating lengthier flow paths. What constitutes evidence that the attachment of GO on the membrane surface takes place through chemical interaction between the amine groups and the oxygenated groups of GO is that the Pe_HE_/Pe_CO2_ separation capacity drops to 2.25 (Figure 12b). In the absence of amine and oxygenated groups, which are the moieties that strongly bound CO_2_, the CO_2_ molecules interact solely with the pure graphitic surface. In this context, surface diffusion has a significant contribution to the overall permeance of CO_2_. Moreover, the He permeance of the Al_2_O_3_ PDA GO membrane is much higher than this of the other two membranes prepared with APTES GO and GLYMO GO (Figure 12c). PDA, due to its bulky conformation, lies solely on the external surface of the Al_2_O_3_ substrate. On the contrary, the smaller APTES and GLYMO molecules can be chemically attached to the pore walls of the Al_2_O_3_ substrate. Therefore, there is a higher possibility that in the APTES and GLYMO-modified membranes, small GO nanosheets are intercalated into the macropores of the substrate, narrowing the pore space and causing significant resistance to He flow. What is also indicative of this conformation is that the He/CH_4_ permeance ratio of the Al_2_O_3_ PDA GO membrane is only 1.34 (Figure 12b) due to CH_4_ passing through the pores of the membrane without any significant resistance and the flux being fortified by the surface diffusion mechanism. On the other hand, in membranes Al_2_O_3_ APTES GO and Al_2_O_3_ GLYMO GO, the He/CH_4_ permeance ratio is higher than 1.8 (Figure 12d), despite the surface diffusion mechanism and, due to the narrowing of the pores, the passage of the bulkier CH_4_ is significantly impeded.

## 5. Conclusions

In this study, porous oligo-layered and multi-layered GO laminate/ceramic composite membranes were successfully prepared via a facile, chemical attachment method, using short- and long-chain molecular linkers with different functional groups. The possible morphological and pore structural characteristics of the deposited laminate relative to the employed linker were examined in-depth by combining spectroscopic and gas permeability measurements that elucidated the gas diffusion mechanism through the porous structure at variable conditions of pressure and temperature. It was found that the chemically anchored laminate sprawls on the top surface of all membranes through alternating domains of high- and low-density assemblies of GO. The gas molecules can be intercalated between the stacked nanosheets by penetrating through imperfections of the GO surface and are governed by the microporous surface diffusion mechanism. The low-density domains are characterized by a looser assemblage of the GO nanosheet stacks and the gas diffusion mechanism spans from Knudsen to Poiseuille, depending on the distance between the adjacent stacks. Hence, the most effective linkers are those concluding to the dominance of high-density segments within the laminate structure.

Spectroscopic analysis of the oligo-layered composite membranes showed that APTES is anchored on the ceramic surface via its alkoxy groups with the aliphatic amine chain being vertically oriented towards the substrate surface and is a less effective linker as compared to GLYMO and PDA. On the contrary, GLYMO was grafted on two sites with its alkoxy and ether group, thus covering the ceramic surface to a larger extent. This conformation concluded with the generation of a higher surface concentration of functional groups available for GO binding. The same behavior holds for the long PDA chains that expanded through the entire ceramic surface. These findings were further confirmed by He permeance measurements at various TMPs and temperatures, where Al_2_O_3_ APTES GO was shown to exhibit Poiseuille flow characteristics. On the contrary, the GLYMO and PDA oligo-layered GO laminate membranes developed on both types of substrates (Al_2_O_3_ and ZrO_2_), along with all the multi-layered ones, presented Knudsen diffusion characteristics, while the dependence of He permeance on temperature unveiled the additional contribution of microporous surface diffusion to the overall flux. This asset was more prominent for the multi-layered GO laminate/ceramic composite membranes. Finally, comparing the He/CO_2_ and He/CH_4_ separation performance with the expected Knudsen selectivity and interpreting the possible mechanisms of microporous diffusion, it was found that the ZrO_2_ GLYMO GO-F membrane was capable of significantly inhibiting the diffusion of the bulkier CH_4_ molecules. As such, it is the membrane characterized by the larger extent of dense domains within the structure of the deposited GO laminate.

## Data Availability

Not applicable.

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
