# Peer review of "Composite GO/Ceramic Membranes Prepared via Chemical Attachment: Characterisation and Gas Permeance Properties"

_membranes, 2022, doi:10.3390/membranes12121181_

Round 1
Reviewer 1 Report
The article discusses an important fundamental issue regarding to fabricating GO membrane on porous ceramic substrates. The characterization and discussion are thorough and meaningful for understanding the functionalizing ceramic support and GO with commonly applied chemical agents. I have comments regarding to details of the manuscript as follows:
1. Line 157, The reference [13] is incorrectly cited. Please double check if there are more mistakes with your reference list.
2. Line 187, “Two types of membranes were developed oligo- and multi-layered GO laminate/ceramic membranes.” Please define oligo- and multi-layered GO membranes as both membranes consist of “multiple” layers of GO.
3. There are a lot of places with “Error! Reference source not found”. Please check and replace proper citation of Figures or other citations.
4. Line 432, “This asset”, the wording is a little weird here.
5. Line 499, “Specifically, the d-distance, which is associated with the size of the GO interlayer galleries, was estimated 8.45 Å, 8.5 Å and 9.57 Å,” d-distance or d-spacing is not direct pore opening number that readers can correlate with. Please add explanation for rough responding pore opening number for easier interpretation.
6. Line 740, “Knudsen and Hagen-Poiseuille mechanisms” The corresponding equations are very helpful here for understanding the discussion.
7. According to the SEM pictures, there are a lot of particles on the membrane surface. Not sure if it is generated during SEM sample preparation or another process. Is it possible that the membrane permeation results are from a defective membrane caused by structural imperfection? Is there preventative steps taken to ensure that the membrane permeation results represent well each targeted membrane structure?
8. I am very confused with the membrane permeation part, it makes sense to certain degree for mechanistic studies. However, I do not think it is necessary to put some much permeation data here. Knudsen diffusion selectivity for most gas pairs is relatively low. Supposedly, a broad pore size range will lead to Knudsen diffusion behavior if adsorption influence is minimal. The discussion does not help generate much meaningful understanding. Practically, it does not make much difference for membrane quality evaluation whether it is above or lower than Knudsen selectivity. I would suggest cutting this part down significantly.
9. For the permeation data figures, it is suggested that keep all figures without outside borders consistently to make the figures neater.
10. Line 1014, “CH4 has not any specific affinity for the oxygenated groups of GO. As such, CH4 permeation benefits solely from the effect of surface diffusion”. I do not think that the reasoning and the conclusion is very solid.
11. The last figure number is wrong.
Author Response
Please see the attachment.
Thank you in advance.

Reviewer 2 Report
In this manuscript, the authors give detailed descriptions from background, materials and methods, to results discussion. The work is of interest. However, there are many shortcomings in the present form. Thus, this manuscript needs to be improved before publication in this journal.
1) Please check if all cited references are relevant to the corresponding content. There are many wrong cited references in the manuscript. For example:
Line 110, “…and textural features of the membrane. [4].”;
Line 115, “…promising [10].”;
Line 118 “…the work of Gu et al. [2]…”;
Line 130 “For instance, Xu et al. [11]…”;
Line 136 “…by Zhang et al. [12],”;
Line 157 “…conducted by Wang et al. [13].”;
Line 344 “…be found in previous publication [7].”
2) Please provide all clear figures with high quality.
3) Scheme I and Scheme II are not indicated in the main text.
4) Al2O3 GLYMO-GO-F and Al2O3 APTES-GO-F stand for multi-layered membranes, is it right? Please indicate in Materials and Methods 2.1.7 and 2.1.8.
5) Figure 4 is not indicated in the main text.
6) Line 533: the authors described “…obtained by SEM analysis (Figures 5 and 8),”. However, Figure 8 not shows the SEM image.
7) Please add the cross-section view of SEM image for Al2O3 APTES-GO.
8) Please reorganize Figure 7a-e and the corresponding caption.
9) Page 28: should be Figure 12.
10) Please make Abstract and Conclusions more concise.
Author Response

(The authors gave the same response as above.)

Round 2
Reviewer 2 Report
The manuscript is significantly improved. I think this manuscript could be published in this journal after revising the following issue.
Why are all references in the main text missing? Please add them.
Author Response
Dear Reviewer,
Thank you for your valuable observation. As we had an unexpected conflict with our word editor (we had a similar conversation via email with the Editor), now in this final version of our revised manuscript, that we will upload, you could see that all references are appropiately mentioned in the main text.
We double-checked our manuscript (word and pdf form) and everything is fine.
Thank you in advance.